# Forecastability of Agricultural Commodity Futures Realised Volatility with Daily Infectious Disease-Related Uncertainty

**Sisa Shiba** [1] , **Goodness C. Aye** [1] , **Rangan Gupta** [1,*] and **Samrat Goswami** [2]

1   Department of Economics, University of Pretoria, Private Bag X20, Hatfield 0028, South Africa
2   Department of Rural Studies, Tripura University, Agartala 799022, Tripura, India
*   Correspondence: rangan.gupta@up.ac.za

**Abstract:** Given the food supply chain disruption from COVID-19 lockdowns around the world, we examine the predictive power of daily infectious diseases-related uncertainty (EMVID) on commodity traded futures within the agricultural bracket, sometimes known as the softs, using the heterogeneous autoregressive realised variance (HAR-RV) model. Considering the short-, medium-, and long-run recursive out-of-sample estimation approach, we estimate daily realised volatility by using intraday data within the 5 min interval for 15 agricultural commodity futures. During the COVID-19 episode, our results indicated that EMVID plays an important role in predicting the future path of agricultural commodity traded futures in the short, medium, and long run, i.e., h = 1, 5, and 22, respectively. According to the MSE-F test, these results are statistically significant. These results contain important implications for investors, portfolio managers, and speculators when faced with investment risk management and strategic asset allocation during infectious disease-related uncertainty.

**Keywords:** commodity futures; infectious disease-related uncertainty; forecasting; realised volatility

**JEL Classification:** C22; Q02

## 1. Introduction

The disruption of food supply chains from COVID-19 lockdowns around the world triggered a tremendous interest in understanding the "safe haven" attribute of agricultural commodity futures (Ji et al. 2020; Sifat et al. 2021; Rubbaniy et al. 2022; Zhang and Wang 2022), raising concerns about the attractiveness of these vehicles in commodity options trading, global supply chain risk management,[1] strategic asset allocations, and regulators' supervision of inflation risk during infectious disease-related uncertainty.

In 2015, the United Nations set 17Sustainable Development Goals (SDGs) that were aimed at improving the standard of living in the world by 2030 (SDSN 2021). Among these SDGs are those of no poverty (SDG1) and zero hunger (SDG2) by 2030. The COVID-19 outbreak imposed the greatest threat to these goals and adversely affected some of the developing progress in achieving them when governments imposed measures such as lockdowns[2] to contain the spread of the virus (Khan et al. 2020). In addition to the lost lives, Béné (2020) emphasised that the main effect of COVID-19 was driven by mobility restrictions by governments, which led to a subsequent loss of income and reduction in purchasing power, especially for low-income individuals and households. The restricted movement between countries (see McBryde et al. 2020) triggered demand and supply shocks (Guerrieri et al. 2022). This threatened food security, the most crucial aspect of sustainable development and economic growth in different parts of the world (Arndt et al. 2020; Mardones et al. 2020; O'Hara and Toussaint 2021). Empirically, approximately 265 million people were affected by food insecurity in 2020, which is a 135 million increase from the COVID-19 outbreak (Food Security Information Network 2020).

The interest of our paper in commodity markets is driven by food security and their more dramatic price fluctuations compared with other financial markets (Hák et al. 2016). If

we think of agricultural commodities, for instance, the production of goods is not uniform throughout the year (de Keizer et al. 2017). Crops, for example, grow in a certain season and are usually harvested a few times a year, and it can often be unpredictable up to a certain point whether the crops will turn out good or bad. The weather conditions have a big effect on these outcomes; however, we may have other unpredictable factors such as pesticides (Tudi et al. 2021). These kinds of fluctuations are a problem for commodity producers, investors, and portfolio managers. However, the COVID-19 outbreak led to high volatility as a result of the high unprecedented uncertainties in the financial market,[3] especially in the commodity markets. Therefore, it is crucial for investors and portfolio managers to mitigate or offset such risk by finding "safe haven" commodity futures during times of infectious diseases.

In times of financial market uncertainties from global crises such as infectious diseases, especially the recent coronavirus pandemic, typically used portfolio risk management strategies are likely to default (Umar and Gubareva 2020; Harjoto et al. 2021). This may result in extreme market volatility because of high trades. More precisely, the disastrous COVID-19 pandemic prompted a high level of uncertainty in the commodity markets although the reaction of such markets differed across countries and traded commodity brackets. For instance, commodity-dependent countries rely heavily on exports and imports as low-and middle-income countries; as a result, they experienced a strong adverse reaction in their markets (Tröster 2020). On the other hand, Borgards et al. (2021) showed that the reaction of agricultural (soft) and metal commodities to the pandemic was minimal except for special treasures such as gold. In addition, Zhang and Hamori (2021) argued that the effects of COVID-19 on the financial markets are more significant compared to other historical shocks such as the 2008 financial crisis, droughts, and floods, although their short-, medium-, and long-run impact is uncertain.

In this context, the objective of our paper is to investigate, for the first time, the predictive ability of daily infectious disease-related uncertainty (EMVID) for agricultural future realised volatilities utilising the heterogeneous autoregressive realised variance (HAR-RV) model. The main attribute of the HAR-RV model is its ability to use volatilities from different time horizons to predict the realised volatility on returns. The model contains the heterogeneous market hypothesis, which states that market participants in their different categories react differently to information flow in the short, medium, and long run (Müller et al. 1997). For example, speculators and traders in the market are more concerned about short-term investments, while investors are more interested in long-term investments. Conventionally, the time-varying volatility is modelled, and the fit is assessed using various generalised autoregressive conditional heteroscedastic (GARCH) models, under which the conditional variance is a deterministic function of model parameters and past data. Alternatively, researchers have also considered stochastic volatility models, where the volatility is a latent variable that follows a stochastic process. These models rely on daily data, and not intraday data as used to obtain RV, which in turn is known to a be more accurate estimate of the latent process of volatility due to the richness of the underlying intraday data (McAleer and Medeiros 2008).

There are a number of studies on the nexus between commodity returns and infectious diseases, especially since the incidence of the COVID-19 pandemic (See Balcilar et al. 2022; Long and Guo 2022; Akyildirim et al. 2022; do Nascimento et al. 2022; Daglis et al. 2020; Umar et al. 2022; Cariappa et al. 2022; Chen et al. 2022; Shruthi and Ramani 2021; Gutierrez et al. 2022; Ayyildiz 2022). However, the current study makes key contributions to the existing literature. First, the focus of existing studies was mainly on the COVID-19 pandemic, while the current study focuses on infectious disease-related uncertainty (EMVID). Secondly, existing studies used daily data for commodity returns, while we use the realised volatility of intraday agricultural commodity futures. Thirdly, relative to existing studies, we analyse the out-of-sample power of EMVID for more (15) agricultural commodity futures (i.e., BO, CC, C, CT, KC, OJ, SB, SM, S, W, FC, LB, LC, LH, and O) (Table A1). The data coverage of uncertainty related to infectious diseases not only covers

the COVID-19 episode, but also includes other infectious diseases such as Ebola, H1N1, H5N1, MERS, or SARS viruses and the recent monkeypox. We use the newspaper-based index by Baker et al. (2020) as a proxy for infectious disease-related uncertainty. The index is derived from the daily equity market volatility (EMV) hosted in the Chicago Board Options Exchange (CBOE) volatility index. This index is robust for a statistical model aimed at forecasting the realised volatility of agricultural commodity futures. Most importantly, the employed intraday data contains information that may result in more accurate and precise estimates and forecasts across different time horizons. Furthermore, this paper contributes to the literature on agricultural commodity futures in that it predicts its realised volatility computed from 5 min intervals utilising the modified heteroscedasticity autoregression model by Corsi (2009). In particular, the basic HAR-RV model is extended by adding the daily infectious disease-related uncertainty (EMVID) variable, and then examining its predictive power on the variables of interest (agricultural commodity futures). Furthermore, we employ recursive out-of-sample predictability of EMVID for the realised volatility of 15 agricultural commodity futures in the short, medium, and long run. In sum, our study is holistic and novel in terms of the wider coverage of infectious disease range, the focus on intraday realised volatility of large number agricultural commodities, the focus on the out-of-sample predictability of EMVID, and the uniqueness of the modified HAR-RV model used, allowing us to conduct short-, medium-, and long-run forecast analysis. To the best of our knowledge, we are not aware of any study that has examined the out-of-sample predictability of EMVID for the intraday volatility of agricultural commodities using the HAR-RV model. This analysis has important implications for portfolio managers in their portfolio diversification possibilities given uncertainties from infectious diseases.

The remaining part of our paper is structured as follows: Section 2 presents the literature review, while Section 3 describes the data and methodology. Section 4 presents the empirical results. Section 5 concludes the paper. Figure A1 presents the data plots for our variables of interest.

## 2. Literature Review

Several studies have been conducted on the infectious disease and financial markets, especially since the incidence of COVID-19 pandemic (see, for example, Salisu and Vo 2020; Salisu et al. 2020; Caggiano et al. 2020; Bouri et al. 2020b; Salisu and Sikiru 2020; Salisu and Adediran 2020; Salisu et al. 2020; Adediran et al. 2021; Liu et al. 2022). However, these studies focused mainly on stock returns. Some studies exist on commodity returns and infectious diseases. For example, using the nonparametric Granger causality-in-quantiles test, Balcilar et al. (2022), assessed the effect of COVID-19 (measured by the news-based sentiment index) on 13 major agricultural commodity prices and price volatility. They employed daily data over 73 months, i.e., from 1 January 2016 to 25 February 2022. Their findings suggest that in both the lower and upper quantile ranges, there is Granger causality from the pandemic to the average commodity prices. Furthermore, COVID-19 sentiment is causal to the price volatility of agricultural commodities in the quantiles above the first quarter.

Long and Guo (2022) analysed the effects of infectious disease equity market volatility and other factors on commodity returns. Results based on time-varying Granger causality test and time-varying parameter vector autoregression with a stochastic volatility model showed that the time-varying effects are significant with mostly positive responses. They also found out that, of the five pandemics (Bird Flu in 1998, SARS in 2003, Swine Flu in 2009, MERS and Ebola in 2014, and COVID-19 in 2019) studied, the recent COVID-19 produced the greatest impact on commodity returns. Furthermore, they showed that the returns of five commodity subcategories, namely, textiles, industry, metals, livestock, and food, were mostly negatively impacted during the sample period, thereby making these commodities not safe haven assets during pandemic risks.

Akyildirim et al. (2022) used panel data regressions and time-varying Granger causality tests to examine whether the spillovers between agricultural commodity returns and

sentiments are influenced by economic and financial uncertainties, including the global COVID-19 pandemic. They found that the agricultural commodity returns and sentiments were significantly influenced by COVID-19-induced uncertainty around the first cycle of the pandemic in 2020. do Nascimento et al. (2022) used the Hurst exponent and multifractal detrended fluctuations analysis, and they found that, during the COVID-19 pandemic (from 1 January 2020 to 25 September 2020), sugar was the most efficient commodity, while pork was the least in the Brazilian agricultural commodity market. Daglis et al. (2020) analysed the impact of COVID-19 pandemic on oats and wheat returns using data from 22 January 2020 to 2 June 2020. Results from the standard VAR model indicated that these markets were affected by COVID-19. Furthermore, these results indicated the out-of-sample forecasting superiority of a model that explicitly incorporates COVID-19 pandemic over the baseline model.

Using data from 1 January 2020 and 30 April 2021, Umar et al. (2022) examined the dynamic return and volatility connectedness for three agricultural commodity indices (softs, grains, and livestock) and the coronavirus media coverage index (MCI). Results based on time-varying parameter vector autoregression showed that dynamic total return and volatility connectedness fluctuated over time, reaching a peak during both the first and the third waves of the COVID-19 pandemic. Cariappa et al. (2022) used time series data from 1 November 2019 to 10 August 2020 in conjunction with survey data to analyse the effect of COVID-19-induced lockdowns on agricultural commodity prices and consumer behaviour in India. Results from an interrupted time series analysis showed a significant rise in the prices of chickpea (4.8%), mung bean (5.2%), and tomato (78.2%), although these reverted immediately after the lockdown. Furthermore, the Kruskal–Wallis test results showed a significant change in consumer behaviour through panic purchases.

Chen et al. (2022) used data from 2019 to 2021 and the Black (1976) model to show how theCOVID-19 pandemic impacted the volatility of Chinese agricultural commodity options more strongly relative to non-agricultural commodities. Using causality in impulse response functions and variance test and daily data from January 2020, Shruthi and Ramani (2021) found that the risk transmission among agricultural commodities was zero. According to Gutierrez et al. (2022), results from a global vector auto regression (GVAR) model revealed that the fall in the oil price may have contributed to the stability of the world grain market in during COVID-19 pandemic and that export restrictions could significantly increase global prices. An asymmetric analysis by Ayyildiz (2022) using the nonlinear autoregressive distributed lag model and data from 11 March 2020 to 11 March 2021 showed that the effect of an increase in the COVID-19 global fear index on agricultural commodity prices was greater than the effect of a decrease.

According to the above, the majority of these studies focused on COVID-19 pandemic while the current study uses an infectious disease uncertainty index that is broad and covers different infectious diseases pandemics. Furthermore, all the studies except Daglis et al. (2020) conducted in-sample predictability analysis, while we conduct an out-of-sample analysis. It is, however, noted that, while Daglis et al. (2020) analysed the out-of-sample predictability of COVID-19 for oats and wheat returns, we analyse the out-of-sample predictability of infectious disease-related uncertainty for the realised volatility of 15 agricultural commodity futures. Hence, we innovate by focusing on volatilities in several agricultural commodities in the futures market.

### 3. Data and Methodology

*3.1. Data*

Data on the realised volatility (RV) of commodity futures were sourced directly from the University of Chicago Booth School of Business Risk Lab under the maintenance of Professor Dacheng Xiu. This series is publicly available at https://dachxiu.chicagobooth. edu/#risklab.com (accessed on 27 April 2022). The highest-frequency available trades were collected and cleaned using the prevalent national best bid and offer (NBBO) that is available every second. The RV estimation procedure was computed using the quasi-

maximum likelihood estimation of volatility (QMLE) from moving average models MA(q), using nonzero returns of transaction prices sampled up to the earliest available frequency for days with at least 12 observations (see Xiu 2010). In choosing the best MA(q), we used the Akaike information criterion. We also employed the 5min RV estimates for our analysis.

The index on dairy infectious disease-related uncertainty (EMVID) is publicly accessible at http://policyuncertainty.com/infectious_EMV.html (accessed on 27 April 2022).This index was developed by Baker et al. (2020) using a newspaper-based infectious disease equity market volatility tracker. In this paper, we use the EMVID data from as early as 22 September 2008 to 27 April 2022 for BO, CC, C, CT, KC, OJ, SB, SM, S, and W RVs, and then from 27 July 2015 to 27 April 2022 for FC, LB, LC, LH, and O RVs (Table A1). EMVID is based on the following four textual analysis terms: E, economic, economy, financial; M, "stock market", equity, equities, "standard and poor"; V: volatility, volatile, uncertain, uncertainty, risky; ID: H1N1, H5N1, MERS, SARS, Ebola pandemic, epidemic, virus, diseases, and coronavirus. In each of the E, M, V, and ID terms, a daily count of at least one term over 3000 US newspaper articles was computed into the EMVID index. On the same day, Baker et al. (2020) multiplicatively rescaled the final series to equal the level of the VIX through the overall EMV index; then, the EMVID index was scaled to total the EMV articles. Our data range varied from the earliest data available to the latest date from our estimation. More interestingly, our data period covers the COVID-19 virus and other economic uncertainties such as the global financial crisis and, more recently, the Russia–Ukraine crisis. Given daily infectious disease-related uncertainty, the EMVID index is the only proxy for uncertainty related to various infectious diseases.

### 3.2. Methodology: Heterogeneous Autoregressive Realised Variance (HAR-RV) Model

To realise the main objective of our paper, we conducted the out-of-sample predictability analysis using the Corsi (2009) HAR-RV model. The key feature of our model is its ability to reproduce the important properties contained in financial data in their respective time intervals while remaining simple (Wang et al. 2019; Gkillas et al. 2020). These properties include fat tails, long memory, multi-scaling behaviour, and self-similarity. The basic HAR-RV model is

$$RV_{t+h} = \beta_0 + \beta_d RV_t + \beta_w RV_{w.t} + \beta_m RV_{m.t} + \varepsilon_{t+h}, \tag{1}$$

where realised volatility (RV) $h$ days ahead is represented by the $h$ index (in our paper, $h = 1, 5,$ and 22); $RV_{w.t}$ represents the average $RV$ from day $t - 6$ to $t - 1$, whereas $RV_{m.t}$ depicts the mean RV from day $t - 22$ to day $t - 6$. We then add the EMVID index to the benchmark HAR-RV model to capture the interest of our paper. $\beta_0$ is a constant, ceteris paribus. $\beta_{d,w \ and \ m}$ are our respective coefficients for the short-, medium-, and long-run RV, while $\varepsilon_{t+h}$ is our error term. The extended HAR-RV model ($\theta$ is the coefficient for daily infectious disease-related uncertainty) is

$$RV_{t+h} = \beta_0 + \beta_d RV_t + \beta_w RV_{w.t} + \beta_m RV_{m.t} + \theta EMVID_t + \varepsilon_{t+h}. \tag{2}$$

## 4. Empirical Results

In this paper, we focus on the out-of-sample predictability of the realised volatility (RV) of commodity traded futures, "the softs"; that is, we access the role that daily infectious disease-related uncertainty (EMVID) plays in predicting the future path of our variables of interest. Campbell (2008) and Bouri et al. (2020a) argued that the best test for any predictive model relies on its out-of-sample performance in terms of any econometric and predictability. We employ an out-of-sample recursive approach from the earliest data available to the latest data for our estimation. The data plots on the variables under investigation in Figure A1 move around the mean with a sharp positive shock that quickly goes back to the mean in the first quarter of the COVID-19 pandemic, especially for our independent variable. Our out-of-sample multiple structural breakpoints tests were determined using the HAR-RV model under the Bai and Perron (2003) test of 1 to M

globally determined breaks and UDMax and WDMax statistics. Table 1 presents the structural breaks.

**Table 1.** Structural breakpoints.

| Date | Symbol | Names |
|---|---|---|
| September 2010 | KC | Coffee "c" futures |
| October 2010 | C, CT, and S | Corn futures, cotton #2 futures, and soybean futures |
| November 2010 | BO and LC | Soybean oil futures and live cattle futures |
| December 2010 | OJ and SB | Orange juice futures and sugar #11 futures |
| March 2011 | CC and SM | Cocoa futures and soybean meal futures |
| October 2011 | W | Wheat futures cbot |
| August 2016 | FC, LB, and O | Feeder cattle futures, lumber futures, and oats futures |
| October 2016 | LH | Lean hogs futures |

As tested by the multiple structural breakpoints test, Table 1 depicts that most agricultural commodity futures experienced multiple structural breaks in 2010. More precisely, corn (C), cotton #2 (CT), and soybean (S) futures experienced a structural breakpoint in October 2010, followed by soybean oil (BO) and live cattle (LC) futures in November 2010. The orange juice (OJ) and sugar #11 (SB) futures had a structural breakpoint in December 2010. In September 2010, the coffee "C" (KC) experienced a breakpoint. Furthermore, cocoa (CC) and soybean meal (SM) futures had a structural breakpoint in March 2011, and wheat futures CBOT (W) experienced a breakpoint in October 2011. Lastly, the feeder cattle (FC), lumber (LB) and oats (O) futures had a structural breakpoint in August 2016, and the lean hogs (LH) futures experienced a breakpoint in October 2016. The important basis of these multiple structural breakpoints involves factors such as food price peaking, reduction in grain stock, low interest rates, and the depreciation of the United States (US) dollar (Headey 2011). Export restrictions, droughts, demand surges, trade shocks, and climate change are among other factors contributing to the global food crisis (see Falkendal et al. 2021; Lieber et al. 2022).

Next, we compute the root-mean-squared forecast errors (RMSFEs) for the benchmark and extended $h = 1$, 5, and 22 HAR-RV models using the above multiple structural breakpoints models. Since our primary aim is to forecast, lower RMSFEs in our recursive out-of-sample estimated from the earliest experienced breakpoint in all the variables of interest would represent a better-performing model. For forecast accuracy, we employ the McCracken (2007) MSE-F test. The out-of-sample forecast gains (FG) were calculated using the following formula:

$$FG = \left( \frac{RMSFE_0}{RMSFE_1} - 1 \right) \times 100, \tag{3}$$

where $RMSFE_0$ denotes the RMSFEs for the benchmark HAR-RV model, while the RMSFEs for the extended HAR-RV model are presented by $RMSFE_1$. Positive or negative FGs indicate the gains or losses in percentage (Equation (3)).

According to our out-of-sample results in Table 2, the highest forecast loss of 0.28% was for the lumber futures (LB), followed by 0.26% forecast loss for soybean oil futures (BO) in the short run ($h = 1$), and then 0.25% in the medium run ($h = 5$) in the BO. This implies that taking the information context of the daily infectious disease-related uncertainty (EMVID) into consideration using the forecast accuracy of the RMSFE metrics within our period of interest, an econometrician can obtain the highest forecast loss of 0.28% for LB ($h = 1$), followed by 0.26% and then 0.25% for BO $h = 1$ and $h = 5$, respectively. Our results also indicate that the coffee "C" ($h = 22$) and oat futures (O) ($h = 5$) remained constant, i.e., there was no forecast gain or loss. However, the lowest forecast loss of 0.01% was in the oat

futures $h = 1$ model, followed by 0.02 for wheat futures CBOT (W) in the $h = 22$ model. This suggests that, considering the information context of uncertainty associated with infectious diseases based on the forecast accuracy of the RMSFE metrics, an econometrician would not be able to obtain any forecast gain or loss for KC ($h = 22$) and O ($h = 5$), but could at least obtain a minimal forecast loss of 0.01% for O ($h = 1$), followed by 0.02% for W ($h = 22$). Considering the whole sample period, these negative FGs also imply that EMVID adds no value in forecasting the realised volatility of our commodity futures. Therefore, the MSE-F test cannot be significant it is a one-sided test associated with whether the unrestricted model does better than the restricted one.

**Table 2.** Full out-of-sample forecasting gains.

| $h$ | $RMSE_0$ | $RMSEE_1$ | FGs | $RMSE_0$ | $RMSEE_1$ | FGs |
|---|---|---|---|---|---|---|
| | Panel 1: BO: 11/18/2010 | | | Panel 2: CC: 3/08/2011 | | |
| 1 | 0.0415 | 0.0416 | −0.2643 | 0.0497 | 0.0497 | −0.0282 |
| 5 | 0.0107 | 0.0107 | −0.2528 | 0.0131 | 0.0131 | −0.0229 |
| 22 | 0.0027 | 0.0027 | −0.0741 | 0.0033 | 0.0033 | −0.0302 |
| | Panel 3: C: 10/29/2010 | | | Panel 4: CT: 10/12/2010 | | |
| 1 | 0.0781 | 0.0781 | −0.0717 | 0.0632 | 0.0633 | −0.1201 |
| 5 | 0.0202 | 0.0202 | −0.0594 | 0.0165 | 0.0165 | −0.0666 |
| 22 | 0.0049 | 0.0049 | −0.0616 | 0.0041 | 0.0041 | −0.0978 |
| | Panel 5: FC: 8/18/2016 | | | Panel 6: KC: 9/14/2010 | | |
| 1 | 0.0503 | 0.0503 | −0.0875 | 0.0578 | 0.0578 | −0.1124 |
| 5 | 0.0129 | 0.0129 | −0.0310 | 0.0152 | 0.0152 | −0.0721 |
| 22 | 0.0018 | 0.0018 | −0.1103 | 0.0038 | 0.0038 | 0.0000 |
| | Panel 7: LB: 8/19/2016 | | | Panel 8: LC: 11/04/2010 | | |
| 1 | 0.1757 | 0.1762 | −0.2798 | 0.0531 | 0.0532 | −0.1937 |
| 5 | 0.0455 | 0.0455 | −0.1383 | 0.0131 | 0.0131 | −0.0915 |
| 22 | 0.0113 | 0.0114 | −0.1674 | 0.0035 | 0.0035 | −0.1442 |
| | Panel 9: LH: 10/11/2016 | | | Panel 10: OJ: 12/29/2010 | | |
| 1 | 0.0740 | 0.0740 | −0.0811 | 0.1239 | 0.1240 | −0.0468 |
| 5 | 0.0184 | 0.0184 | −0.0760 | 0.0324 | 0.0324 | −0.0309 |
| 22 | 0.0049 | 0.0049 | −0.1233 | 0.0078 | 0.0078 | −0.0385 |
| | Panel 11: O: 8/17/2016 | | | Panel 12: SB: 12/30/2010 | | |
| 1 | 0.1394 | 0.1394 | −0.0065 | 0.0570 | 0.0570 | −0.0526 |
| 5 | 0.0366 | 0.0366 | −0.0027 | 0.0148 | 0.0148 | −0.0271 |
| 22 | 0.0087 | 0.0087 | −0.0345 | 0.0037 | 0.0037 | −0.0540 |
| | Panel 13: SM: 3/21/2011 | | | Panel 11: S: 10/21/2010 | | |
| 1 | 0.0536 | 0.0536 | −0.0280 | 0.0461 | 0.0461 | −0.0390 |
| 5 | 0.0139 | 0.0139 | −0.0359 | 0.0120 | 0.0120 | −0.0334 |
| 22 | 0.0034 | 0.0034 | −0.0291 | 0.0029 | 0.0029 | −0.0344 |
| | Panel 15: W: 10/03/2011 | | | | | |
| 1 | 0.0683 | 0.0684 | −0.0702 | | | |
| 5 | 0.0183 | 0.0184 | −0.0436 | | | |
| 22 | 0.0046 | 0.0046 | −0.0219 | | | |

Note: $FG = \left( \frac{RMSFE_0}{RMSFE_1} - 1 \right) \times 100$ was the formula used to calculate the forecasting gains (FG), where $RMSFE_0$ stands for the root-mean-squared forecast errors ($RMSFE_S$) for the benchmark model, and $RMSFE_1$ represents the $RMSFE_S$ for the extended HAR-RV model. $RV_{t+h} = \beta_0 + \beta_d RV_t + \beta_w RV_{w,t} + \beta_m RV_{m,t} + \varepsilon_{t+h}$ is the equation for the benchmark HAR-RV model, and $RV_{t+h} = \beta_0 + \beta_d RV_t + \beta_w RV_{w,t} + \beta_m RV_{m,t} + \theta EMVID_t + \varepsilon_{t+h}$ is the equation for the extended HAR-RV model. RV depicts the daily realised volatility for agricultural commodity futures, whilethe daily infectious disease-related uncertainty is shown by EMVID.

Across all economic agents, the interest in searching for "safe haven" vehicles given infectious disease-related uncertainty was triggered by the COVID-19 outbreak; therefore,

it is crucial to assess the impact of EMVID within the COVID-19 period. As the primary purpose of this paper, we conducted a recursive out-of-sample estimation from January 2020 to the earliest period of our estimation and computed the in-sample period including the same number of observations. That is, we performed in- and out-of-sample observations. This period incorporates all phases of COVID-19. Within the COVID-19 episode, our results in Table 3 depict that the cocoa futures (CC) had the highest FG of 265.12% in the $h = 22$ model, followed by 119.38% for oats futures in the $h = 22$ model, and then 91.40% for sugar #11 futures ($h = 1$). This implies that, by incorporating the information context of infectious disease-related uncertainty such as COVID-19 using the forecast accuracy of the RMSFE metrics, an econometrician could acquire the highest FG of 265.12% for CC ($h = 22$), followed by 119.37% for O ($h = 22$), and then 91.40 for SB ($h = 1$). Furthermore, within the same episode, the lowest forecast gains of 0.70%, 1.49%, and 1.68%were evident in the SM ($h = 5$), SB ($h = 22$), and SM ($h = 1$), respectively. This means that, considering COVID-19-related uncertainty and the forecast accuracy RMSFE metrics, an econometrician could obtain the lowest FGs of 0.70% in SM ($h = 5$), followed by 1.49% for SB ($h = 22$), and then 1.68% for SM ($h = 1$). According to the MSE-F critical values,[4] these results were statistically significant at a 1% level of significance except for BO in the $h = 1$ and $h = 5$ models. Most importantly, the results of our out-of-sample in the COVID-19 episode indicate the extent to which trade openness can be affected by a national shutdown given infectious diseases. Specifically, the supply shock triggered food insecurity; as a result, there was a high willingness to hedge against such risks.

**Table 3.** COVID-19 episode out-of-sample forecasting gains.

| $h$ | $RMSE_0$ | $RMSEE_1$ | FGs | $RMSE_0$ | $RMSEE_1$ | FGs |
|---|---|---|---|---|---|---|
| | | Panel 1: BO: 01/02/2019 | | | Panel 2: CC: 01/02/2019 | |
| 1 | 0.0593 | 0.0668 | −11.2464 | 0.0683 | 0.0441 | 54.9028 *** |
| 5 | 0.0151 | 0.0164 | −8.0490 | 0.0167 | 0.0113 | 48.3345 *** |
| 22 | 0.0039 | 0.0036 | 8.8284 *** | 0.0107 | 0.0029 | 265.1210 *** |
| | | Panel 3: C: 01/02/2019 | | | Panel 4: CT: 01/02/2019 | |
| 1 | 0.1225 | 0.0743 | 64.7963 *** | 0.1141 | 0.0673 | 69.4641 *** |
| 5 | 0.0336 | 0.0195 | 72.1085 *** | 0.0195 | 0.0172 | 13.8156 *** |
| 22 | 0.0058 | 0.0049 | 18.4884 *** | 0.0046 | 0.0044 | 4.2970 *** |
| | | Panel 5: FC: 01/02/2019 | | | Panel 6: KC: 01/02/2019 | |
| 1 | 0.0619 | 0.0523 | 18.4753 *** | 0.0787 | 0.0703 | 11.9315 *** |
| 5 | 0.0203 | 0.0169 | 19.9965 *** | 0.0245 | 0.0182 | 34.4523 *** |
| 22 | 0.0026 | 0.0023 | 11.3804 *** | 0.0050 | 0.0047 | 6.8548 *** |
| | | Panel 7: LB: 01/02/2019 | | | Panel 8: LC: 01/02/2019 | |
| 1 | 0.2823 | 0.2492 | 13.3014 *** | 0.1038 | 0.0737 | 40.7548 *** |
| 5 | 0.0861 | 0.0641 | 34.3329 *** | 0.0239 | 0.0179 | 33.5645 *** |
| 22 | 0.0165 | 0.0161 | 2.4484 *** | 0.0050 | 0.0048 | 2.9724 *** |
| | | Panel 9: LH: 01/02/2019 | | | Panel 10: OJ: 01/02/2019 | |
| 1 | 0.0977 | 0.0871 | 12.2664 *** | 0.1552 | 0.1310 | 18.5122 *** |
| 5 | 0.0302 | 0.0211 | 43.3042 *** | 0.0408 | 0.0335 | 21.8513 *** |
| 22 | 0.0063 | 0.0059 | 6.7586 *** | 0.0124 | 0.0079 | 56.6002 *** |
| | | Panel 11: O:01/02/2019 | | | Panel 12: SB:01/02/2019 | |
| 1 | 0.2343 | 0.1449 | 61.6987 *** | 0.0990 | 0.0517 | 91.4020 *** |
| 5 | 0.0378 | 0.0370 | 2.2160 *** | 0.0199 | 0.0131 | 52.1473 *** |
| 22 | 0.0197 | 0.0090 | 119.3689 *** | 0.0034 | 0.0034 | 1.4784 *** |

**Table 3.** *Cont.*

| *h* | $RMSE_0$ | $RMSEE_1$ | FGs | $RMSE_0$ | $RMSEE_1$ | FGs |
|---|---|---|---|---|---|---|
| | Panel 13: SM: 01/02/2019 | | | Panel 14: S: 01/02/2019 | | |
| 1 | 0.0517 | 0.0508 | 1.6788 *** | 0.0623 | 0.0458 | 36.0493 *** |
| 5 | 0.0133 | 0.0133 | 0.7018 *** | 0.0152 | 0.0118 | 27.9527 *** |
| 22 | 0.0060 | 0.0035 | 72.6407 *** | 0.0030 | 0.0030 | 0.0000 *** |
| | Panel 15: W: 01/02/2019 | | | | | |
| 1 | 0.1612 | 0.0935 | 72.4723 *** | | | |
| 5 | 0.0429 | 0.0257 | 66.8597 *** | | | |
| 22 | 0.0084 | 0.0063 | 32.7129 *** | | | |

Note: $FG = \left( \frac{RMSFE_0}{RMSFE_1} - 1 \right) \times 100$ was the formula used to calculate the forecasting gains (FG), where $RMSFE_0$ stands for the root-mean-squared forecast errors ($RMSFE_S$) for the benchmark model, and $RMSFE_1$ represents the $RMSFE_S$ for the extended HAR-RV model. $RV_{t+h} = \beta_0 + \beta_d RV_t + \beta_w RV_{w,t} + \beta_m RV_{m,t} + \varepsilon_{t+h}$ is the equation for the benchmark HAR-RV model, and $RV_{t+h} = \beta_0 + \beta_d RV_t + \beta_w RV_{w,t} + \beta_m RV_{m,t} + \theta EMVID_t + \varepsilon_{t+h}$ is the equation for the extended HAR-RV model. RV depicts the daily realised volatility for agricultural commodity futures, while the daily infectious disease-related uncertainty is shown by EMVID. The MSE-F test denotes the level of significance at the 1% level, as represented by ***.

## 5. Conclusions

Given food insecurity problems as a result of the COVID-19 lockdowns around the world, we investigated the forecasting ability of daily infectious disease-related uncertainty (EMVID) with respect to the realised volatility of agricultural commodity traded futures. We employed the heterogeneous autoregressive realised variance (HAR-RV) model by Corsi (2009) on 15 commodity traded futures. Considering our recursive out-of-sample estimation approach in the short, medium, and long run within the COVID-19 episode, it is evident that cocoa futures (CC) had the highest FG of 265.12% in the long run (*h* = 22), followed by oat futures (O) with 119.38% FG in *h* = 22, and then 91.40% FG for sugar #11 (SB) in the short run (*h* = 1). This implies that, considering the information context of the forecasting accuracy for RMSFE metrics within the COVID-19 period, an econometrician could obtain the highest FG of 265.12% in CC *h* = 22, followed by 119.38% for O *h* = 22, and then 91.40% for SB *h* = 1. An econometrician could also obtain the lowest FG of 0.70%, followed by 1.49% and 1.68% in SM *h* = 5, SB *h* = 22, and SM *h* = 1, respectively.

Our results within the COVID-19 episode suggest that EMVID plays an important role in predicting the future path of agricultural commodity futures. These findings have important implications for portfolio managers and investors in their search for safe investment or diversification options in the financial market. These results are robust as suggested by McCracken's (2007) MSE-F test. The COVID-19 pandemic is the worst crisis the world had seen; therefore, there are limited related studies and measures or indices for COVID-19. Furthermore, the pandemic already aggravated existing food insecurity problems and other global challenges; hence, we cannot blame the volatility of this asset class under review solely on the pandemic. In the future, we expect to extend our study to other brackets of agricultural commodities such as those in the metal bracket.

**Author Contributions:** Conceptualization, S.S., G.C.A., R.G. and S.G.; methodology, S.S.; validation, S.S., G.C.A., R.G. and S.G.; formal analysis S.S., G.C.A., R.G. and S.G.; writing—original draft preparation, S.S., G.C.A., R.G. and S.G.; writing—review and editing, S.S., G.C.A., R.G. and S.G. All authors have read and agreed to the published version of the manuscript.

**Funding:** This research received no external funding.

**Data Availability Statement:** Data are available under request from the authors, but the raw data is publicly available as stated in the data segment.

**Acknowledgments:** We thank the anonymous referees; any remaining errors are solely our responsibility.

**Conflicts of Interest:** The authors declare no conflict of interest.

## Appendix A

**Table A1.** Selected variables, acronyms, and sample coverage.

| Symbol | Future Index | Sample Period |
| --- | --- | --- |
| 1. BO | Soybean oil futures | 22 September 2008–27 April 2022 |
| 2. CC | Cocoa futures | 22 September 2008–27 April 2022 |
| 3. C | Corn futures | 22 September 2008–27 April 2022 |
| 4. CT | Cotton no.2 futures | 22 September 2008–27 April 2022 |
| 5. FC | Feeder cattle futures | 27 September 2015–27 April 2022 |
| 6. KC | Coffee c futures | 22 September 2008–27 April 2022 |
| 7. LB | Lumber futures | 27 July 2015–27 April 2022 |
| 8. LC | Live cattle futures | 27 July 2015–27 April 2022 |
| 9. LH | Lean hogs futures | 27 July 2015–27 April 2022 |
| 10. OJ | Orange juice futures | 22 September 2008–27 April 2022 |
| 11. O | Oat futures | 27 July 2015–27 April 2022 |
| 12. SB | Sugar #11 futures | 22 September 2008–27 April 2022 |
| 13. SM | Soybean meal futures | 22 September 2008–27 April 2022 |
| 14. S | Soybean futures | 22 September 2008–27 April 2022 |
| 15. W | Wheat futures CBOT | 22 September 2008–27 April 2022 |

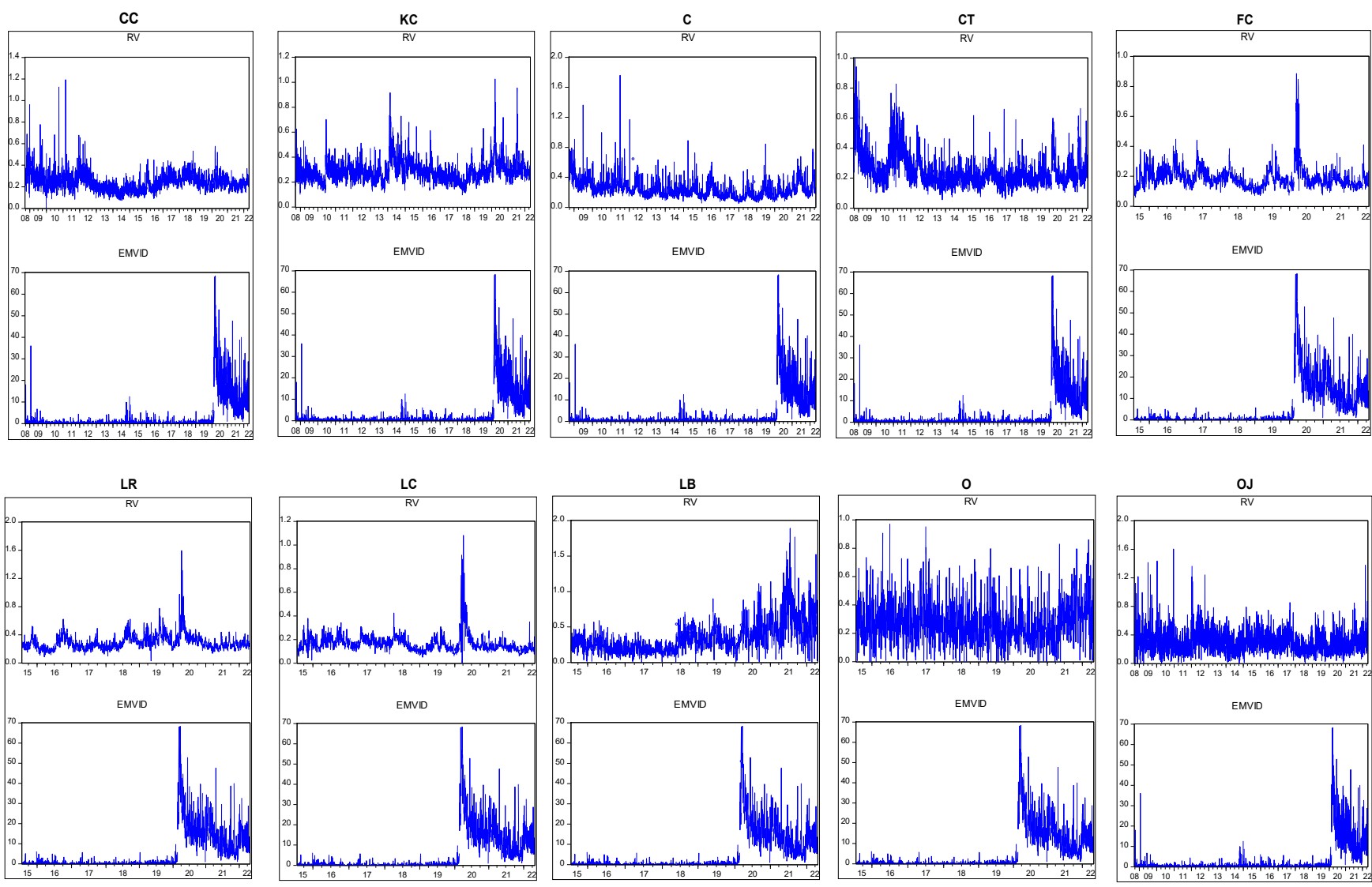

**Figure A1.** *Cont.*

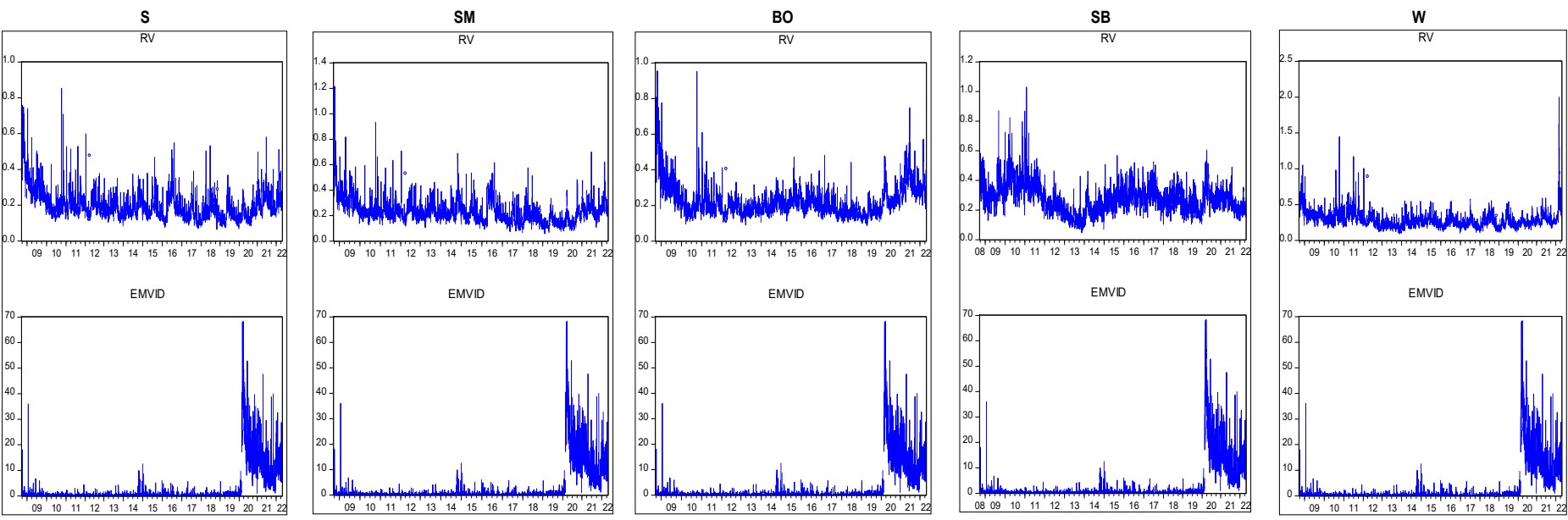

**Figure A1.** Data plots. Notes: The realised volatility of the agricultural commodity futures is represented by RV. The newspaper-based uncertainty index related to infectious disease is represented by EMVID.

## Notes

[1]   The profitability of businesses heavily depends on risk management strategies to hedge futures cash flow uncertainty.

[2]   Lockdowns reduced the movement of goods and services and even brought some to zero, i.e., movements of imports and exports.

[3]   Liao et al. (2018) noted the following three channels through which the fluctuation in the financial market can impact commodity prices: macro-economy reflection channel, financial market information transmission channel, and market sentiment contagion channel.

[4]   MSE-F critical values: 3.584, 1.548, and 0.751.

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
