# Peer review of "Forecastability of Agricultural Commodity Futures Realised Volatility with Daily Infectious Disease-Related Uncertainty"

_jrfm, doi:10.3390/jrfm15110525_

Round 1

Reviewer 1 Report

The manuscript reported " Forecastability of Agricultural Commodity Futures Realised Volatility with Daily Infectious Disease-Related Uncertainty". Authors need to clarify some issues to make their work effective. These are as follows:

1-    The authors need to revise the abstract by including research problem, objective and main results and research recommendations that is practically attainable.

1-    The existing related researches should be better summarized, and the research novelty should be emphasized in the introduction section.

1-  The results should be extended to support the proposed approach effectively.

Author Response

We thank the anonymous referees and any remaining errors are solely ours.

Reviewer 2 Report

The article is interesting and presents the study of the disease of public importance in the international trade of selected agricultural commodity futures. Please add in Results a short analysis of figure 1. 

Author Response

(The authors gave the same response as above.)

Reviewer 3 Report

This paper aims to investigate for the first time the predictive ability of daily infectious disease-related uncertainty (EMVID) for agricultural future realised volatilities utilising the heterogeneous autoregressive realised variance (HARRV) model.

However, I have some comments as follows:

1. Need to revise the abstract, by highlighting the main findings of this study.

2. The symbols in the equation (1) and (2) should be defined clearly.

3.In the conclusion section, paragraph 1 to 3 is more to introduction and repeating. Please revise the conclusion by highlighting the main findings of this study.

4. Please include the limitation of this study in the conclusion section.

5. Check and ensure the format of the paper is correct.

Author Response

(The authors gave the same response as above.)
